# Observational operators for dual polarimetric radars in variational data assimilation systems (PolRad VAR v1.0)

Takuya Kawabata[1], Thomas Schwitalla[2], Ahoro Adachi1, Hans-Stefan Bauer[2], Volker Wulfmeyer[2], Nobuhiro Nagumo[1], and Hiroshi Yamauchi[3]

[1]Meteorological Research Institute, Japan Meteorological Agency, Tsukuba, 3050052, Japan
[2]Institute of Physics and Meteorology, University of Hohenheim, Stuttgart, 70599, Germany
[3]Japan Meteorological Agency, Tokyo, 1008122, Japan

*Correspondence to*: Takuya Kawabata (tkawabat@mri-jma.go.jp)

**Abstract.** We implemented two observational operators for dual polarimetric radars in two variational data assimilation systems: WRF Var, the Weather Research and Forecasting Model variational data assimilation system, and NHM-4DVAR, the nonhydrostatic variational data assimilation system for the Japan Meteorological Agency nonhydrostatic model. The operators consist of a space interpolator, two types of variable converters as well as their linearized and transposed (adjoint) operators. The space interpolator takes account of the effects of radar-beam broadening in both vertical and horizontal directions and climatological beam bending. The first variable converter emulates polarimetric parameters with model prognostic variables and includes attenuation effects, and the second one derives rainwater content from the observed polarimetric parameter (specific differential phase). We developed linearized and adjoint operators for the space interpolator and variable converters and then assessed whether the linearity of the linearized operators and the accuracy of the adjoint operators were good enough for implementation in variational systems. The results of a simple assimilation experiment showed good agreement between assimilation results and observations with respect to reflectivity and specific differential phase but not with respect to differential reflectivity.

## 1 Introduction

The Weather Research and Forecasting Model (WRF; Skamarock et al., 2008) is a widely used numerical weather model that was developed as a community model, and WRF Var, its data assimilation system (Barker et al., 2012), provides initial conditions for the model. NHM-4DVAR is a nonhydrostatic 4DVar system for the Japan Meteorological Agency nonhydrostatic model (JMANHM; Saito et al., 2012 that functions at storm scale (Kawabata et al., 2007, 2014a). Many remote sensing data are available for NHM-4DVAR, such as slant total delay, zenith total delay, and precipitable water vapour observed by global navigation satellite systems (GNSS; Kawabata et al., 2008); conventional radar data, including directly assimilated reflectivity data (Kawabata et al., 2011); and Doppler lidar data (Kawabata et al., 2014b). Because data assimilation associates observations with model fields, to make use of advanced observations, data assimilation methods need to be continuously developed and implemented into variational data assimilation systems.

Observations obtained by dual polarimetric radars are utilized by operational systems at many meteorological and hydrological operation centres (e.g., in the United States, France, Germany and Japan) to improve the accuracy of quantitative precipitation estimation (QPE). These radars provide polarimetric parameters, including the horizontally polarized reflectivity factor ($Z_H$), the vertically polarized reflectivity factor ($Z_V$), differential reflectivity ($Z_{DR}$), and the specific differential phase ($K_{DP}$). Many QPE methods that use these parameters have been proposed (e.g., Jameson, 1991; Jameson and Caylor, 1994; Ryzhkov and Zrnić, 1995; Anagnostou et al., 2008; Kim et al., 2010; Ryzhkov et al., 2014; Adachi et al., 2015). Because QPE methods using dual polarimetric radar parameters are expected to be better than methods using single polarization radar data, we developed assimilation methods for dual polarimetric radar observations for both WRF Var and NHM-4DVAR. The objective of our study was thus to improve QPE, which was discussed in Bauer et al. (2015) in the context of a data assimilation with high resolution and rapid update cycle, and quantitative rainfall forecasts (QPF) through the use of better analysis fields obtained by the assimilation of dual polarimetric radar observations.

We chose an emulator (Zhang et al., 2001) and an estimator (Bringi and Chandrasekar, 2001) to use as forward operators after evaluating their accuracy (Kawabata et al., 2018). In addition, because both WRF Var and NHM-4DVAR consider only perturbations to rainwater in their tangent and adjoint models, our operators also deal only with rainwater and exclude ice particles. Although both this emulator (Jung et al., 2008a, 2008b) and this estimator (Yokota et al., 2016) have been used previously as observational operators in ensemble Kalman filter data assimilation systems, to our knowledge, our study is their first implementation in variational assimilation systems. We call the current version of the operators as PolRad VAR v1.0.

The first author has mainly contributed to the WRF Var version of these operators developed over the rapid update WRF 3DVAR system at the University of Hohenheim, Germany (see, e.g., Schwitalla et al., 2011; Schwitalla and Wulfmeyer, 2014; Bauer et al. 2015), and to the version for NHM-4DVAR at Meteorological Research Institute, Japan Meteorological Agency. .

The scope of this paper is to provide the technical information on the observational operators and some evaluation results to help the users understand theoretical and practical aspects of the operators. The forward operators (space interpolator and variable converters) and their linearized (tangent linear) and transposed (adjoint) operators are described in Sect. 2. Section 3 describes setup options of the observational operators, Sect. 4 presents verification and assimilation test results, and Sect. 5 is a summary.

## 2 Observational operators

In variational data assimilation systems, a cost function is defined and then iteratively minimized until its gradient becomes zero. The cost function and its gradient are defined as

$$J(\mathbf{x}) = \frac{1}{2}(\mathbf{x} - \mathbf{x}^b)^{\mathrm{T}}\mathbf{B}^{-1}(\mathbf{x} - \mathbf{x}^b) + \frac{1}{2}(H(\mathbf{x}) - \mathbf{y})^{\mathrm{T}}\mathbf{R}^{-1}(H(\mathbf{x}) - \mathbf{y}), \tag{1}$$

$$\nabla J(\mathbf{x}) = \mathbf{B}^{-1}(\mathbf{x} - \mathbf{x}^b) + \mathbf{H}^{\mathrm{T}}\mathbf{R}^{-1}(H(\mathbf{x}) - \mathbf{y}), \tag{2}$$

where T denotes the transpose of a matrix; $\mathbf{x}$, $\mathbf{x}^b$, and $\mathbf{y}$ are model fields, first-guess fields, and observations, respectively; and $H(\mathrm{x})$, $\mathbf{H}$, and $\mathbf{H}^T$ represent the observational operators, their linearized operators (tangent linear), and their transposed (adjoint) operators, respectively. The observational operators work as variable converters ($H_v$) from model fields $\mathbf{x}$ to observational values related to observations $\mathbf{y}$, and as space interpolators ($H_s$) from model to observational space as follows:

$$H(\mathbf{x}) = H_s H_v(\mathbf{x}). \tag{3}$$

We developed two types of variable converters, a single space interpolator, and their tangent linear and adjoint operators. Both WRF Var and NHM-4DVAR consider only perturbation to the mixing ratio of the rainwater and not that to its number density in the tangent linear and adjoint models. However, in the tangent and adjoint operators described here (Sect. 2.2), non-perturbed number density of rainwater is included. This variable is initialized to zero at the beginning or end of the operators, and this effect is directly considered in the cost functions of WRF Var and NHM-4DVAR, whereas its gradient is indirectly considered through perturbations of the mixing ratios of rainwater, water vapour and other variables like temperature and pressure.

It is recommended that users of WRF Var run the system with CLOUD_CV (required) and the CV7 (optional) switches. The former adds mixing ratios of rainwater to the default control variable set (Wang et al., 2013), and the latter replaces the control variables of stream function and velocity potential with momentum control variables to improve the performance of WRF simulations at high horizontal resolution (Sun et al., 2016). With these selections, the control variables in WRF Var are almost the same as those in NHM-4DVAR (Kawabata et al., 2011).

## 2.1 Variable converters

### 2.1.1 Model variables to polarimetric parameters (FIT)

Among the many numerical precipitation scheme options (e.g., single-moment scheme, large-scale condensation scheme) for WRF and JMANHM, we chose double-moment schemes (WRF, Morrison et al., 2009; JMANHM, Hashimoto, 2008) for our observational operators because such schemes predict both the number density ($N_r$; m$^{-3}$) and the mixing ratio ($Q_r$; kg kg$^{-1}$) of rainwater, whereas single-moment schemes predict only $Q_r$. Therefore, two of three unknown parameters in the drop size distribution (DSD) function are detected by the schemes. Following Morrison et al. (2009), the DSD function is given by

$$N(D) = N_0 D^\mu exp(-\Lambda D), \tag{4}$$

where $D$ (mm) is the raindrop diameter, $N_0$ (mm$^{-1}$ m$^{-3}$) is the intercept parameter, $\mu$ is the shape parameter, and $\Lambda$ (mm$^{-1}$) is the slope parameter. $\Lambda$ is given by

$$\Lambda = \left(\frac{\pi \rho_w N_r}{10^3 \rho_a Q_r}\right)^{\frac{1}{3}}, \tag{5}$$

where $\rho_w$ is the density of water (997 kg m$^{-3}$ in this study) and $\rho_a$ is air density (kg m$^{-3}$), a model diagnostic variable. $\rho_a$ and $N_0$ are given by

$$\rho_a = \frac{p}{RT(1 + 0.61q_v)},\tag{6}$$

$$N_0 = N_r\Lambda,\tag{7}$$

where $p$ is atmospheric pressure (Pa), $R$ is the gas constant, $T$ is temperature (K), and $q_v$ is the mixing ratio of water vapour (kg kg$^{-1}$).

5    In our study, the remaining unknown parameter $\mu$ is fixed at zero, and $N(D)$ is based on bulk sampling, the minimum and maximum values of $D$ are set to 0.05 mm and 5 mm, respectively.

Because in the rainwater prognostic variables, raindrops are assumed to be spherical in both WRF and JMANHM, we introduce the axis ratio of a raindrop, which is polynomial to $D$ (Brandes et al., 2002, 2005), as follows:

$$r = 0.9951 + 2.51 \times 10^{-2}D - 3.644 \times 10^{-2}D^2$$

$$+5.303 \times 10^{-3}D^3 - 2.492 \times 10^{-4}D^4.\tag{8}$$

Radar observations are derived from measurements of the scattering of electromagnetic waves by raindrops. The first converter is based on fitting functions that relate equivolume diameters $D$ to scattering amplitude (Zhang et al., 2001). The backscattering amplitudes are represented by a power law function as follows:

$$\left|S_{h,v}(D)\right| = \alpha_{h,v}D^{\beta_{h,v}},\tag{9}$$

where the coefficients $\alpha_{h,v}$ and $\beta_{h,v}$ are determined by fitting $D$ to the backscattering amplitudes $\left|S_{h,v}\right|$ calculated by the T-matrix method (Mishchenko et al., 1996). The difference between the horizontal and vertical forward scattering amplitudes is defined as

$$\mathrm{Re}\big(f_h(D) - f_v(D)\big) = \alpha_k D^{\beta_k},\tag{10}$$

where $f_h(D)$ and $f_v(D)$ represent the horizontal and vertical forward scattering amplitudes, and $\alpha_k$ and $\beta_k$ are determined by

the fitting. Zhang et al. (2001) proposed fitting functions for S-band radars, and Kawabata et al. (2018) derived new fitting parameters for C-band radars. Following Zhang et al. (2001), horizontal (H) and vertical (V) reflectivity factors are

$$Z_{\mathrm{H,V}} = \frac{4\lambda^4}{\pi^4|K_w|^2}\big(\alpha_{h,v}^2 N_0\Lambda^{-(2\beta_{h,v}+1)}\Gamma(2\beta_{h,v} + 1)\big),\tag{11}$$

where $\lambda$ (m) is the radar wavelength; $K_w$ is a constant, defined as $K_w = (\varepsilon - 1)/(\varepsilon + 2)$, where $\varepsilon$ is the complex dielectric constant of water estimated as a function of wavelength and temperature (Sadiku, 1985); and $\Gamma$ represents the Gamma

function. The horizontal reflectivity $Z_{\mathrm{H}}$ is converted to conventional reflectivity $Z_h$ (dBZ) by

$$Z_h = 10\log_{10}(Z_{\mathrm{H}}),\tag{12}$$

and $Z_{\mathrm{DR}}$ (dB) is defined as

$$Z_{\mathrm{DR}} = 10\log_{10}\left(Z_{\mathrm{H}}\big/Z_{\mathrm{v}}\right) = Z_h - Z_v.\tag{13}$$

$K_{\mathrm{DP}}$ ($^\circ$ km$^{-1}$) is defined as

$$K_{\mathrm{DP}} = \frac{180\lambda}{\pi}N_0\alpha_k\Lambda^{-(\beta_k+1)}\Gamma(\beta_k + 1).\tag{14}$$

The attenuation effects are calculated as follows:

$$Z_h^{att}(x) = Z_h(x) - 2\int_0^x A_H(s)ds,\tag{15}$$

$$Z_{DR}^{att}(x) = Z_{DR}(x) - 2\int_0^x A_{DP}(s)ds,\tag{16}$$

where $Z_h^{att}$ and $Z_{DR}^{att}$ represent attenuated $Z_h$ and $Z_{DR}$, respectively. $A_H$ and $A_{DP}$ are the specific attenuation (dB km$^{-1}$) and the specific differential attenuation (dB km$^{-1}$), respectively, defined as

$$A_H = \alpha_H K_{DP}{}^{\beta_H},\tag{17}$$

$$A_{DP} = \alpha_d K_{DP}{}^{\beta_d},\tag{18}$$

The values of the coefficients $\alpha_h,\ \alpha_v,\ \alpha_k,\ \alpha_H,$ and $\alpha_d$ and $\beta_h, \beta_v, \beta_k, \beta_H,$ and $\beta_d$ for C-band in these equations are listed in Table 1. Hereafter, this converter is called FIT.

FIT is also applicable for X- and S-bands by replacing the coefficients. Although we already prepared the coefficients for all bands in the source codes, the users should carefully investigate their validity.

### 2.1.2 Observations of polarimetric parameters to model variables (KD)

The second converter (hereafter KD) converts observed $K_{DP}$ to rainwater content ($Q_{rain}$) according to the following relation:

$$Qrain = 3.565\left(\frac{K_{DP}}{f}\right)^{0.77},\tag{19}$$

where $f$ (GHz) is the radar frequency and the power law coefficients are from Bringi and Chandrasekar (2001). $Q_{rain}$ in the model is defined as $Q_{rain} = Q_r\rho_a$ (kg m$^{-3}$). Note that Eq. (19) is applicable for not only C-band but also X- and S-bands by putting their frequencies in $f$.Eqs. (4)-(19) follow Kawabata et al. (2018), and we put the equations with different order in this manuscript for the readers' convenience to understand the flow of implementations of the forward, tangent linear and adjoint codes

### 2.2 Tangent linear and adjoint operators

### 2.2.1 Tangent linear and adjoint operators of FIT

Because only $p$, $T$, and $q_v$ are perturbed in WRF Var and NHM-4DVAR, the linearized form of Eq. (6) is

$$\Delta\rho_a = \frac{\Delta p}{RT(1 + 0.61q_v)} - \frac{p\Delta T}{RT^2(1 + 0.61q_v)} - \frac{0.61p\Delta q_v}{RT(1 + 0.61q_v)^2},\tag{20}$$

and the perturbations of $\Lambda$ and $N_0$ are given as

$$\Delta\Lambda = \frac{1}{3}\Lambda(-\Delta Q_r Q_r^{-1} - \Delta\rho_a\rho_a^{-1}),\tag{21}$$

$$\Delta N_0 = N_r\Delta\Lambda,\tag{22}$$

where $\Delta Q_r$ and $\Delta N_0$ are perturbations of the mixing ratio and number density of rainwater, respectively. Note that the perturbation of $N_r$ is not considered in the adjoint model (see Sect. 2). Thus, the perturbations of $Z_{H,V}$, $Z_{DR}$, and $K_{DP}$ are represented as

$$\Delta Z_{H,V} = \frac{4\lambda^4}{\pi^4 |K_w|^2}\left(\alpha_{h,v}^2 \Gamma(2\beta_{h,v}+1)(\Delta N_0 \Lambda^{-(2\beta_{h,v}+1)} - (2\beta_{h,v}+1)\Delta\Lambda N_0 \Lambda^{-(2\beta_{h,v}+2)})\right),$$

(23)

$$\Delta Z_{DR} = \Delta Z_h - \Delta Z_v,$$
(24)

$$\Delta K_{DP} = \frac{180\lambda}{\pi}\alpha_k \Gamma(\beta_k+1)(\Delta N_0 \Lambda^{-(\beta_k+1)} - (\beta_k+1)\Delta\Lambda N_0 \Lambda^{-(\beta_k+2)}).$$
(25)

Finally, the perturbations of $A_H$ and $A_{DP}$ are

$$\Delta A_H = \alpha_H \beta_H \Delta K_{DP} K_{DP}^{\beta_H-1},$$
(26)

$$\Delta A_{DP} = \alpha_d \beta_d \Delta K_{DP} K_{DP}^{\beta_d-1}.$$
(27)

The adjoint operators are represented by the transposed form of Eqs. (20)–(27), that is, $(tangent\ linear)^T$. As an example, the adjoint of Eq. (27) is

$$\Delta K_{DP} = \Delta K_{DP} + \alpha_d \beta_d K_{DP}^{\beta_d-1}\Delta A_{DP}.$$
(28)

## 2.2.2 Tangent linear and adjoint operators of KD

Because $K_{DP}$ in Eq. (19) is an observed value, it is not necessary to linearize the equation. However, the equation that relates $Q_{rain}$ to $Q_r$ (Sect. 2.1.2) needs to be linearized as follows:

$$\Delta Q_{rain} = \Delta Q_r \rho_a + Q_r \Delta \rho_a.$$
(29)

The transposed form of this equation is used for the adjoint model (see Sect. 2.2.1).

## 2.3 Space interpolator

Space interpolators in data assimilation systems map the model space to the observational space according to the representativeness of the observations. In the case of radar data, the effect of beam broadening stands for the representativeness, typically for a beam width of approximately $1.0°$. The broadening is characterized by a Gaussian distribution orthogonal to the direction to the radar beam. Most previous studies (e.g., Seko et al., 2004; Wattrelot et al., 2014), except Zeng et al. (2016), consider only vertical beam broadening, because numerical models have horizontal grid spacings of several kilometres, whereas they have vertical grid spacings in the lower troposphere of less than one kilometre. However, data assimilation systems must have sub-kilometre horizontal grid spacings as well (e.g., Kawabata et al. 2014a, Miyoshi et al., 2016) so that the space interpolators can take account of horizontal beam broadening. In addition, several phased array radars recently

deployed in Japan have different beam widths in the vertical and horizontal directions. Our operator thus considers beam broadening in both the vertical and horizontal directions.

In addition, it is important for the space interpolator to include beam-bending effects, which depend on atmospheric conditions. In this study, the bending is determined by considering the climatological vertical gradient of the refractive index of the atmosphere in accordance with the effective earth radius model (Doviak and Zrnić, 1993), following Haase and Crewell (2000), who showed statistically that the climatological refractive index is close to the actual refractive index at elevation angles higher than 1°, instead of by considering the actual atmospheric conditions, although Zeng et al. (2014) developed an excellent radar simulator that considers the actual refractivity of the atmosphere.

Remote sensing observations usually have higher spatial resolutions than the model grid spacings. To avoid correlations of the observational errors in such high-resolution data, it is necessary either to thin the data or to use "super observations". In this study, we chose the super observation method, in which observations are averaged over each model grid cell. Super observation methods also have the advantage that they remove undesirable fluctuations associated with sub-grid-scale phenomena, the assimilation of which makes the numerical model unnecessarily noisy (e.g., Seko et al., 2004; Zhang et al., 2009).

First, we calculated the path of the centre of the radar beam in the model domain, including its elevation, azimuth, and bending angles (Fig. 1a). Once sufficient data are included within a model grid cell, they are averaged and mapped onto an interpolation point along the radar beam (IP in Fig. 1). This value at this point is a "super observation", and it is compared with the modelled value, which was interpolated by using Gaussian weights (Fig. 1b). Moreover, we also developed the tangent and adjoint codes of the space interpolator.

## 3 Setup options

The operators are controlled by the namelist ("namelist.polradar") as follows:

&name_obs o_dir='/home/usr/datadir', o_stn(1)='OFT', o_stn(2)='TUR', icnv=0/.

Here, 'o_dir' is the directory for the input observational data; 'o_stn' indicates the station names of radar sites, where 'max_stn', the number of names, is set in 'da_setup_obs_structures_polradar.inc' in WRF Var and in 'obs_dual_pol.f90:' in NHM-4DVAR; and 'icnv' is a switch for the selection of the observational operator, where "0" and "5" mean FIT and KD, respectively.

In addition, a file that defines for each radar areas where the beam is blocked by topography, named 'beam_block_rate_${radar_site}.dat', must be supplied by the user. This file is made by another program and should be prepared before the assimilation.

## 4 Results

### 4.1 Verification of the tangent linear and adjoint operators of FIT

In this section, we examine the linearity of only the FIT variable converter; it is not necessary to examine the linearity of the KD converter because of the intrinsic linearity of Eq. (19). We evaluated the linearity of FIT by performing a Taylor expansion.

5    If the original equation is given as

$$\mathbf{y} = H(\mathbf{x}), \tag{30}$$

then the linearized equation is defined as

$$\delta \mathbf{y} = \mathbf{H}\delta \mathbf{x}. \tag{31}$$

If the linear equation is derived with no errors, the following Taylor expansion of Eq. (31)

$$\frac{|H(\mathbf{x} + \alpha \delta \mathbf{x}) - H(\mathbf{x})|}{|\alpha||\mathbf{H}\delta \mathbf{x}|} = 1 + O(\alpha) \tag{32}$$

10    should be accurate within the rounding error of the computer. The results for $Z_H$, $Z_V$, and $K_{DP}$ in Eqs. (11) and (14) are 1.00 when $\alpha$ is $10^{-7}$ to $10^{-15}$.

    Regarding the adjoint operator, we evaluated the following equation:

$$(\mathbf{H}\delta \mathbf{x})^{\mathbf{T}}(\mathbf{H}\delta \mathbf{x}) = \delta \mathbf{x}^{\mathbf{T}}[\mathbf{H}^{\mathbf{T}}(\mathbf{H}\delta \mathbf{x})], \tag{33}$$

15    where the left-hand side of Eq. (33) is calculated using the tangent linear operator, and on the right-hand side, the output variables of the tangent linear operator are input into the adjoint operator. This equation must be accurate within the rounding error. In FIT, the difference between the left- and right-hand sides was $-8.215650382 \times 10^{-15}$, which we consider accurate enough.

### 4.2 Actual data assimilation test

20

We conducted two simple data assimilation tests. Observational errors of $Z_h$, $Z_{DR}$, $K_{DP}$, and $Q_{rain}$, which were determined after the statistical examination (Kawabata et al. 2018), were 15.0 dBZ, which is the same with Kawabata et al. (2011), 2.0 dB, 4.0° $km^{-1}$, and 4 g $m^{-3}$, respectively. These errors are homogeneous in space, which means observational error covariances are diagonal.

25    The first one was done using NHM-4DVAR with actual radar data from the C-band dual polarimetric radar at the Meteorological Research Institute in Tsukuba, Japan (Yamauchi et al., 2012; Adachi et al., 2013). In this experiment, both of radial velocity data and polarimetric parameters of $Z_H$, $Z_{DR}$, and $K_{DP}$ were assimilated in FIT, and radial velocity and $Q_{rain}$ derived from $K_{DP}$ was assimilated in KD. The assimilation window was from 2100 to 2105 UTC 23 June 2014, a day on which intense hail fell in Tokyo, Japan. The horizontal resolution of NHM-4DVAR was 2 km and the length of assimilation window

30    was 5 min, eleven PPI data from 0.5° to 4.8° elevations with the azimuth resolution at 0.7° and the range resolution of 150 m were assimilated. Each PPI data was assimilated at exact observation time as far as the time interval of NHM-4DVAR (10

s in this case) permits. The background errors were described in Kawabata et al. (2007) and (2011). Analysis (KD and FIT) and observational (OBS) fields of $Z_h$, $Z_{DR}$, and $K_{DP}$ are shown in Fig. 2, which displays the whole assimilation domain. Although there was no rain region in the first-guess field (FG; Fig. 2d), $Z_h$ in KD was comparable to that in OBS from the standpoint of rainfall distribution and intensity, but $Z_h$ in FIT covered a much smaller area than it did in OBS. This smaller

5 coverage may be due to nonlinearity in FIT. In KD, we can see quite small values of $K_{DP}$ (Fig. 2f), but good agreement with OBS in its horizontal distribution, while $Z_h$ looks better than $K_{DP}$. $K_{DP}$ values were smaller in both KD and FIT than in OBS. This result is similar to that of a statistical analysis performed by Kawabata et al. (2018). In contrast, $Z_{DR}$ values in KD and FIT were larger than OBS over large areas. This result implies that the calculation of the axis ratio of raindrops (Eq. 8) may need modification, because in the FG field, $Z_{DR}$ values and coverage were already too large, in comparison with those of OBS.

The second one was done using WRF 3DVAR with actual radar data from the DWD radar network (Helmert et al. 2014) for the same case with "Case 1" described in Kawabata et al. (2018). The horizontal resolution of WRF 3DVAR was 2 km, and polarimetric parameters and rain water content in single PPI data by Offenthal radar was assimilated (see Kawabata et al. 2018 for detailed information on the observation). The background errors were calculated with ensemble simulations by WRF initialized by ECMWF analysis using the "gen_be" tool compiled in WRFDA. Observational errors were the same with the

first case. From the increments of polarimetric parameters (Fig. 3), although quite small impacts are seen, similar patterns are recognized in both methods and larger impact of $Z_h$ and $Z_{DR}$ were produced in FIT and KD, respectively.

In both cases, the radial velocity data were assimilated as the same method with Sun and Crook (1997).

## 5 Summary

We implemented two variable converters for polarimetric radars in the WRF variational data assimilation system (WRF Var)

and the JMANHM data assimilation system (NHM-4DVAR). FIT simulates polarimetric parameters using a double moment cloud microphysics scheme, and KD estimates rainwater contents with the observed specific differential phase. Both of FIT and KD are applicable for not only C-band but also X- and S-bands. The advantage of FIT over KD is that it includes theoretically precise formulations for both the mixing ratio and number density of rainwater, as well as attenuation effects, whereas KD has advantages due to its linear formulation and small computational cost.

These operators work in conjunction with an advanced space interpolator, which considers 1) beam broadening in three dimensions, 2) different beam widths in vertical and horizontal directions, 3) the climatological beam-bending effect. The interpolator also simulated attenuation effects.

Tangent and adjoint operators of the two variable converters and the space interpolator were developed and implemented along with the forward operators. In a simple data assimilation experiment, we succeeded in assimilating actual polarimetric

observations and obtained reasonable results with both the FIT and KD operators, except for $Z_{DR}$. However, our results show a need for further improvements of the $K_{DP}$ and $Z_{DR}$ estimates. It would be possible to overcome the weaknesses of the $Z_h$ distributions in FIT and FG through assimilation–forecast cycles and/or by adding other types of observation data, such as

conventional observations, Doppler (water vapour) lidar data, and water vapour data observed by GNSS. Furthermore, it is necessary to improve quality controls (QC) for polarimetric parameters, although the same QCs were applied as described in Kawabata et al. (2018), the impact of axis ratio (Eq. (8)) and observational errors on assimilations will be investigated, and it is necessary to estimate more appropriate observational errors (e.g., Wulfmeyer et al. 2016). These challenges would improve QPE and QPF with the current forms of the operators.

*Code data availability*

Since PolRad VAR v1.0 for NHM-4DVAR belongs to the Meteorological Research Institute of the Japan Meteorological Agency and is not publicly available, any researchers interested in the code are encouraged to contact the corresponding author and sign a contract for license to get the code. PolRad VAR v1.0 for WRF Var is currently being implemented into the community version of WRF Var and will be accessible at the WRF repository (http://www2.mmm.ucar.edu/wrf/users/downloads.html) in the near future. Any researchers interested in the current form of the code can get it from the corresponding author via e-mail.

**Acknowledgements**

We are grateful to the WRF development and support team. This study was partly supported by the Catchments As Organized System (CAOS) project funded by the German Research Foundation (FOR 1598); the Japanese Ministry of Education, Culture, Sports, Science and Technology "Advancement of meteorological and global environmental predictions utilizing observational Big Data" project; the Japan Science and Technology Agency (CREST) "Innovating Big Data Assimilation technology for revolutionizing very-short-range severe weather prediction" project (JPMJCR1312); and Grants-in-Aid for Scientific Research No. 16H04054, "Study of optimum perturbation methods for ensemble data assimilation", and No. 17H02962, "Study on uncertainties in convection initiations and developments using a particle filter", from the Japan Society for the Promotion of Science.

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

Table 1. Values of coefficients $\alpha$ and $\beta$. $\alpha_{h,v,k}$ and $\beta_{h,v,k}$ in Eqs. (9) and (10) and $\alpha_{H,d}$ in Eqs. (17) and (18) are from Kawabata et al. (2018), and $\beta_{H,d}$ in Eqs. (17) and (18) are from Bringi and Chandrasekar (2001).

Figure 1. Schematic diagrams of (a) a super observation and (b) the space interpolator. Boxes represent model grid cells, and the red box indicates the grid cell in which the super observation is defined. The cross marks represent the interpolation point (IP). In (b), the grey curve indicates the Gaussian weights at various grid points (black circles); the solid black line shows the beam propagation; and the dashed lines illustrate beam broadening.

Figure 2. Horizontal distributions of polarimetric parameters of, from left to right, observations (OBS), assimilation results by KD and FIT with NHM-4DVAR, and the first-guess field (FG) at 2104 UTC on 23 June 2014. (a)–(d) $Z_h$; (e)–(h) $K_{DP}$; (i)–(l) $Z_{DR}$.

Figure 3. Horizontal distributions of differences of polarimetric parameters between assimilation results by KD and FIT with WRF 3DVAR and observations at 1100 UTC on 14 August 2014.

Table 1. Values of coefficients $\alpha$ and $\beta$. $\alpha_{h,v,k}$ and $\beta_{h,v,k}$ in Eqs. (9) and (10) and $\alpha_{H,d}$ in Eqs. (17) and (18) are from Kawabata et al. (2018), and $\beta_{H,d}$ in Eqs. (17) and (18) are from Bringi and Chandrasekar (2001).

| (subscript) | $h$ | $v$ | $k$ | $H$ | $d$ |
|---|---|---|---|---|---|
| $\alpha$ | 0.0016 | 0.0017 | $2.36 \times 10^{-5}$ | 0.073 | 0.013 |
| $\beta$ | 2.98 | 2.77 | 5.36 | 0.99 | 1.23 |

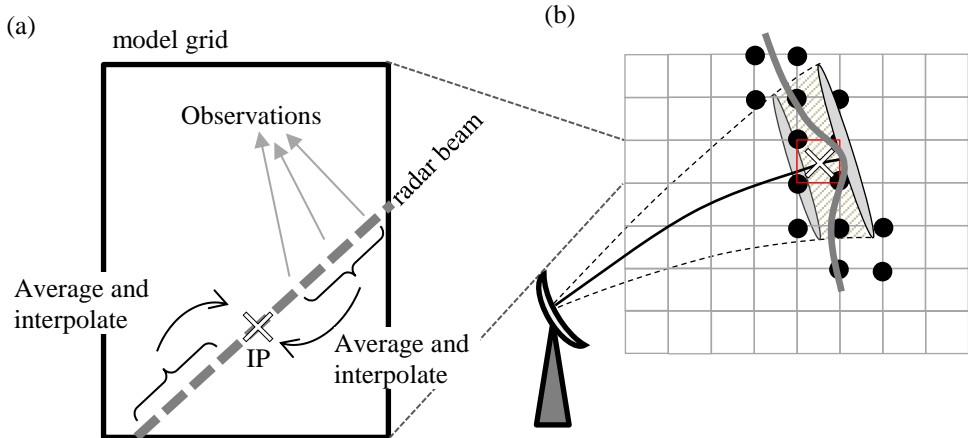

**Figure 1.** Schematic diagrams of (**a**) a super observation and (**b**) the space interpolator. Boxes represent model grid cells, and the red box indicates the grid cell in which the super observation is defined. The cross marks represent the interpolation point (IP). In (b), the grey curve indicates the Gaussian weights at various grid points (black circles); the solid black line shows the beam propagation; and the dashed lines illustrate beam broadening.

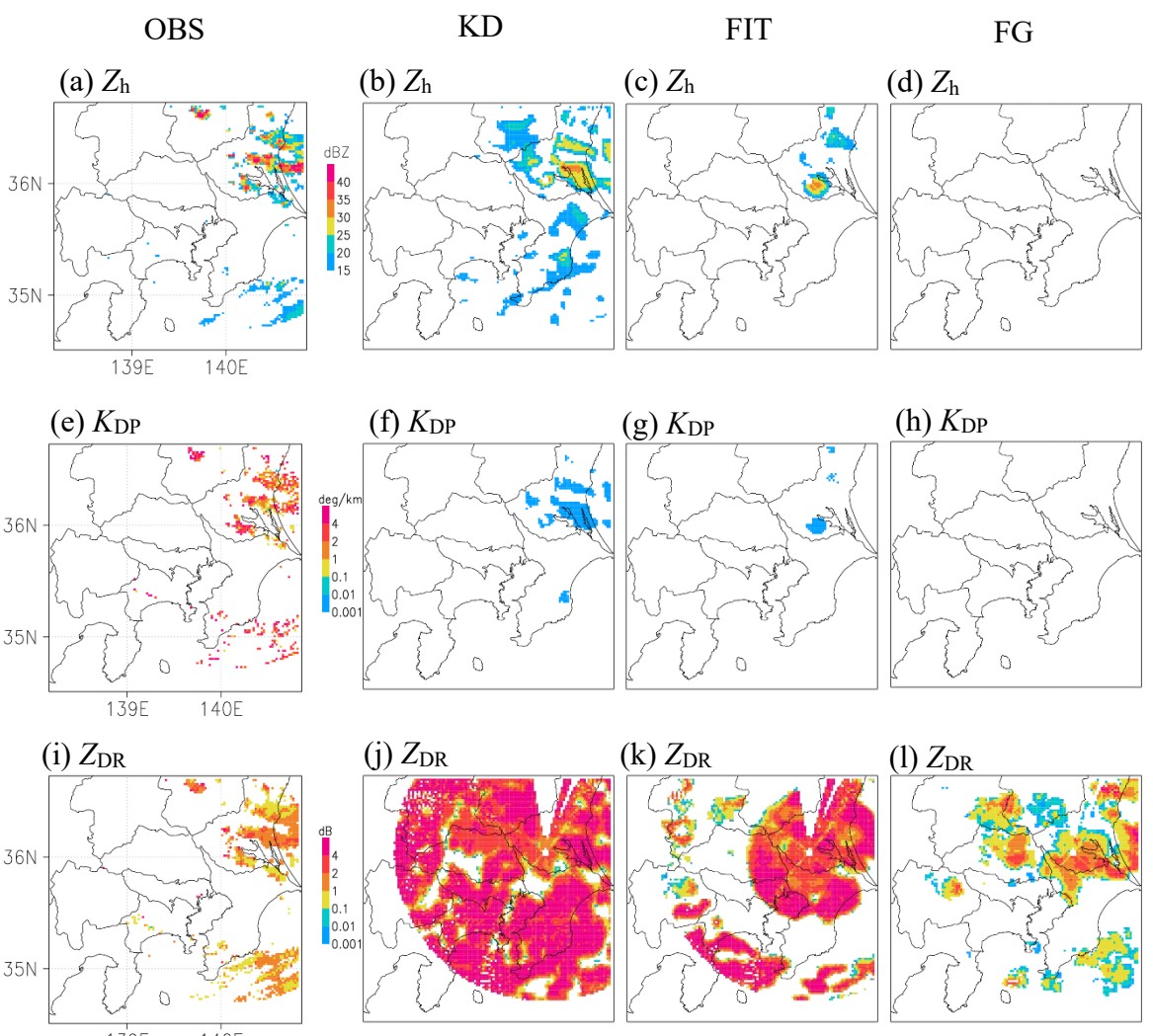

**Figure 2.** Horizontal distributions of polarimetric parameters of, from left to right, observations (OBS), assimilation results by KD and FIT with NHM-4DVAR, and the first-guess field (FG) at 2104 UTC on 23 June 2014. (**a**)–(**d**) $Z_h$; (**e**)–(**h**) $K_{DP}$; (**i**)–(**l**) $Z_{DR}$.

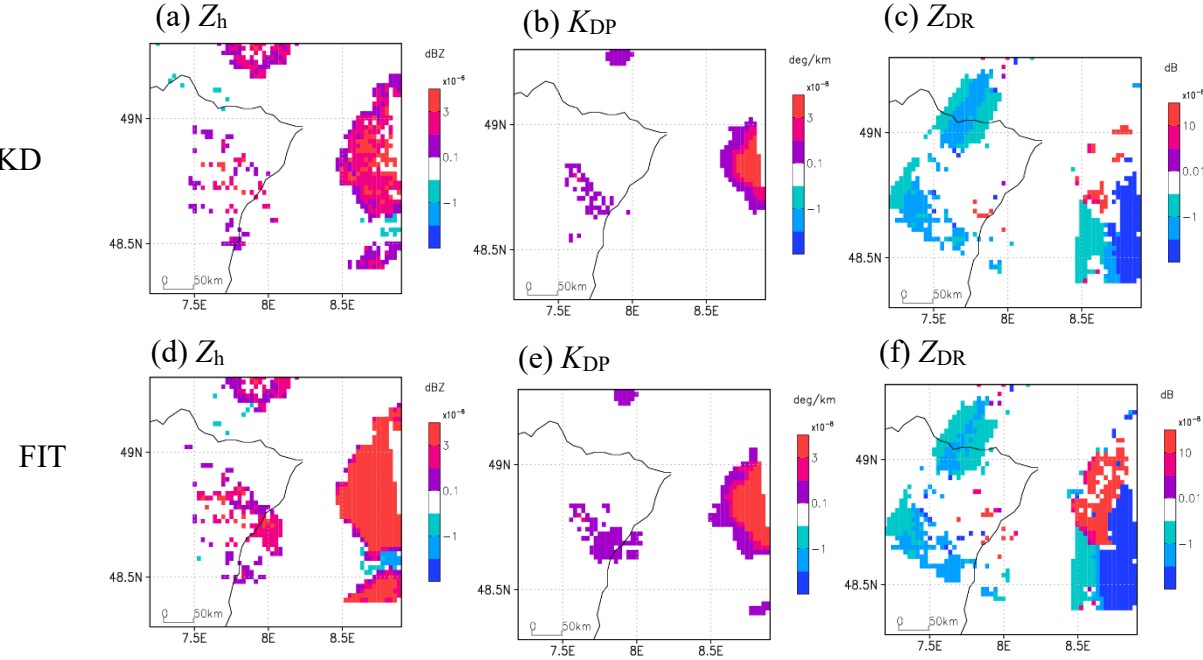

**Figure 3.** Horizontal distributions of differences of polarimetric parameters between assimilation results by KD and FIT with WRF 3DVAR and observations at 1100 UTC on 14 August 2014.