# Peer review of "Observational operators for dual polarimetric radars in variational data assimilation systems"

_Geoscientific Model Development, 2018_

## Short Comment (SC1) · 4 Apr 2018

Dear authors,

in my role as Executive editor of GMD, I would like to bring to your attention our Editorial version 1.1: http://www.geosci-model-dev.net/8/3487/2015/gmd-8-3487-2015.html This highlights some requirements of papers published in GMD, which is also available on the GMD website in the 'Manuscript Types' section: http://www.geoscientific-model-development.net/submission/manuscript_types.html In particular, please note that for your paper, the following requirement has not been met in the Discussions paper:

- "The main paper must give the model name and version number (or other unique

identifier) in the title."

Therefore provide the name and the version number of the newly developed operators in the title of your revised manuscript. Note, that a name and a version number are important to identify these specific operators.

As explained in https://www.geoscientific-model-development.net/about/manuscript_types.html GMD is encouraging authors to upload the program code of models (including relevant data sets) as supplement or make the code and data of the exact model version described in the paper accessible through a DOI (digital object identifier). In case your institution does not provide the possibility to make electronic data accessible through a DOI you may consider other providers (eg. zenodo.org of CERN) to create a DOI. Please note that in the code accessibility section you can still point the reader to how to obtain the newest version. If for some reason the code and/or data cannot be made available in this form (e.g. only via e-mail contact) the "Code Availability" section need to clearly state the reasons for why access is restricted (e.g. licensing reasons). Especially, please note, that it is not enough, that the code will be available in the future. It must be available now and the exact version of the code published in this article needs to be made available.

Yours, Astrid Kerkweg

---

## Author Comment (AC1) · 10 Apr 2018

We appreciate useful comments for improving our manuscript. We respond to the comments as follows:

[Comment by Dr. Astrid Kerkweg] Therefore provide the name and the version number of the newly developed operators in the title of your revised manuscript.

[Response by the authors] The title "Observational operators for dual polarimetric radars in variational data assimilation systems" will be changed to "Observational operators for dual polarimetric radars in variational data assimilation systems (PolRad VAR v1.0)". In addition, the sentense "We call the current version of the operators as PolRad VAR v1.0." will be added to the end of P2L15.

[Figure]

[Comment by Dr. Astrid Kerkweg] Please note that in the code accessibility section you can still point the reader to how to obtain the newest version. If for some reason the code and/or data cannot be made available in this form (e.g. only via e-mail contact) the "Code Availability" section need to clearly state the reasons for why access is restricted (e.g. licensing reasons). Especially, please note, that it is not enough, that the code will be available in the future. It must be available now and the exact version of the code published in this article needs to be made available.

[Response by the authors] The code availability section in page 9 will be modified to clarify more how to get the code as follows: "Since PolRad VAR v1.0 for NHM-4DVAR belongs to the Meteorological Research Institute of the Japan Meteorological Agency and is not publicly available, any researchers interested in the code are encouraged to contact the corresponding author and sign a contract for license to get the code. Although PolRad VAR v1.0 for WRF Var is currently being implemented into the community version of WRF Var and will be accessible at the WRF repository (http://www2.mmm.ucar.edu/wrf/users/downloads.html) in the near future, any researchers interested in the current form of the code can get it from the corresponding author via e-mail."
* * *

---

## Referee Comment (RC1) · J. Sun (Referee) · 13 Apr 2018

General Comments This paper presented two forward operators for dual-pol radar data assimilation and compared their effect on analysis. The operators were developed for WRF VAR and NHM-4DVAR. The former system has been widely used by the community and the latter has been used operationally by JMA for a number of years. This is the first paper, to my knowledge, that has compared the two types operators - converting model variables to observation variables and converting observational variables to model variables. The work is significant and approach is appropriate. The subject is clearly presented and the result is convincing. The presentation of the modeled operators can be easily followed and they should be able to be reproduced either by following the paper or collaborate with the authors. Below lists my specific comments that the

authors may want to clarify.

Specific Comments 1. Page 2, line 9: "The objective of our study was thus to improve QPE and ..." How can you improve QPE by assimilating the dual-pol data with the developed operators? 2. Page 2, line 16: "...our study is their first implementation in variational assimilation systems". Note that Li and Mecikalski (2010) implemented an dual-pol operator in WRF Var similar to your KD. 3. Page 4, line 18: Change "the fitting" to "a statistical fitting". 4. Page 8, section 4.2 and Figure 2: What DA system did you use to produce the results in Figure 2? If you developed the operators for the two systems, it should be natural to show the analyses from both systems, right? Page 8, line 16: These errors are quite large. Have you tried to use smaller errors? Page 9, line 8: "....reasonable results with both the FIT and KD operators". This statement is not accurate. The result from KD is reasonable and clearly better than that from FIT for Zh. The result of Zdr from FIT has some characteristics of the observed Zdr but not that from KD. The Kdp from both FIT and KD differ quite significantly from the observation. Can you speculate why the Kdp is so poorly represented? From the Eq (19), Qr and Kdp have a quite simple relationship but why Zh is rather reasonalble but not Kdp?

---

## Referee Comment (RC2) · S. Sugimoto (Referee) · 2 May 2018

General comments:

In this manuscript, the authors propose forward operators together with their tangent linear (TL) and adjoint (AD) operators for observational data from dual polarimetric radar. This investigation was done by following their previous work on developing forward operators (Kawabata et al. 2018, JMSJ). To my knowledge, the development of theoretically-based TL and AD codes for polarimetric parameters is the first attempt. Operators were derived in appropriate manners, and they could be useful in a framework of variational data assimilation. The results, however, show the low performance in a simple assimilation experiment. The authors have to revise this manuscript mainly

the relevant section of this 4DVar assimilation experiment before acceptance for publication. I think the authors need to reconsider the experimental configurations with additional consideration of the results, especially on the performance of FIT and the errors found in differential reflectivity field. My comments are addressed below.

1. The authors had better mention clearly the range of application in abstract and summary. For example, operators can be applicable to C-band radar data (possibly S-band radar using findings of previous works). Another work, however, is needed to perform a statistical fitting of results simulated by a numerical radar simulator in applying to a radar system with a shorter wavelength (e.g., X-band radar). Besides, a dataset of beam-filling (effected by the ground) is required. In terms of a mesoscale model, the use of a two moment microphysical scheme is assumed.

2. The descriptions in Section 2.1 are quite similar to the description found in the previous work of Kawabata et al. (2018). The authors should explain the essence of FIT concisely for the readers to understand that the forward operators have been already proposed in another work. Please revise Section 2.1 carefully avoid double posting. This revise may be reflected to title.

3. The authors have to describe experimental configurations (Section 4.2) in detail, including the domain, the grid spacing of the mesoscale model used, 4DVar timeline, radar data configuration (e.g., resolutions, the number of elevation angles) at least. Which mesoscale model is used, WRF or NHM? In terms of timeline, are several PPI data in one volume scan assimilated at the precise scanning timing during assimilation window? How about a method for preparing the background error?

4. Observational errors are quite large, and, especially, the error for horizontal radar reflectivity seems to be unrealistic. A smaller error of radar reflectivity should be used. The authors may show the data to support the set-up of errors. The sensitivity of errors to the results should be discussed.

5. Why not radial velocities (rv) assimilated in case of KD? Anyway, hail is associated

with the event investigated. How does the authors consider the fall speed of hydrometeor in assimilating rv data?

6. In Figure 2, non-zero differential reflectivity (Zdr) is calculated in non-stormy areas of retrieved and background fields, regardless of the type of operator. A possible reason regarding the axis ratio does not make sense to me. The authors should mention logically the reasons together with plotting smaller radar reflectivity Zh (< 15 dBZ). Assimilation of radar data (Zh, Zdr, Kdp) with quite weak echoes (e.g., clear air echo) is not appropriate in this framework. In operators, quite small Qr can lead to huge contribution in the perturbations of lamda, N0, and Zh (Zv). Therefore, the use of the minimum thresholds for Zh and Qr may remedy the low performance for the retrieval of Zdr, if the authors does not set the thresholds.

7. Although FIT is theoretically more precise than KD, FIT shows the lower performance than KD. The authors jump to conclusions too quickly by regarding the nonlinearity in FIT as the low performance. In the background Zh and Kdp, the mesoscale model cannot resolve convections at all. One possible situation is that it is too dry in the background water vapor field. If so, I guess the adjustment of humidity is needed before assimilation to retrieve larger Zh and Kdp.

Specific and minor comments:

1. (Page 2) Why do the authors address on quantitative precipitation estimation (QPE)? I think QPE is out of the main topic of this manuscript.

2. (Page 2 Line7) Change "is" to "are".

3. (Pages 2 and 3) Operators proposed consider the relations between variables concerning rain water and radar observables. I feel something wrong with the mention of "cloud water". Can C-band radar observe cloud water?

3. (Pages 2 and 3) Please check if the WRFDA "(WRF-Var" mentioned in the manuscript) deals with the "perturbation" of rainwater mixing ratio, not dealing with

"the total part".

4. (Page 4 Line 6) Change "proportional" to "polynomial".

5. (Page 8 Line 17) Essentially, is Kdp assimilate in KD?

6. (Page 9 Lines 8 & 9) The performance found in a simple assimilation test is far from a successful level.

---

## Author Comment (AC2) · 23 May 2018

Dear Dr. Juanzhen Sun,

The authors deeply appreciate you for your useful comments. Our sincere responses are shown as follows and will be presented in the final form of our manuscript.

Sincerely, Takuya Kawabata on behalf of the authors

[Comment] 1. Page 2, line 9: "The objective of our study was thus to improve QPE and ..." How can you improve QPE by assimilating the dual-pol data with the developed operators?

[Response] As shown in Bauer et al. (2015), high resolution DA systems with rapid

update cycle is able to produce rainfall distributions comparable to actual observations. We added the following words at P2L9.

", which was discussed in Bauer et al. (2015) in the context of a data assimilation with high resolution and rapid update cycle,"

[Comment] 2. Page 2, line 16: "...our study is their first implementation in variational assimilation systems". Note that Li and Mecikalski (2010) implemented an dual-pol operator in WRF Var similar to your KD.

[Response] We removed the expression regarding KD and modify the sentence to "Although this emulator (Jung et al., 2008a, 2008b) has been used previously as observational operators in ensemble Kalman filter data assimilation systems, to our knowledge, our study is their first implementation in variational assimilation systems." (P2L15-16)

[Comment] 3. Page 4, line 18: Change "the fitting" to "a statistical fitting".

[Response] Thank you for your suggestion. We changed it like as it.

[Comment] 4. Page 8, section 4.2 and Figure 2: What DA system did you use to produce the results in Figure 2? If you developed the operators for the two systems, it should be natural to show the analyses from both systems, right?

[Response] We used NHM-4DVAR for Fig. 2. Figures by WRF DA were added as Figure 3 and explanations were displayed in the 3rd paragraph of Section 4.2

[Comment] Page 8, line 16: These errors are quite large. Have you tried to use smaller errors?

[Response] The values of the observational errors were determined after statistical investigations (Kawabata et al. 2018) but they were adopted conservatively (larger than the statistics). (P8L21-22)) The sensitivity of the errors will be examined over scientific explores.

[Comment] Page 9, line 8: "....reasonable results with both the FIT and KD operators".

This statement is not accurate. The result from KD is reasonable and clearly better than that from FIT for Zh.

[Response] Since this statement is from the comparison with the first guess field (FG), which catches no rain it is clear that the results are better than FG. However, since we agree with you that ZDR from both methods are poor, we will add "except for ZDR" at the end of the sentence (P9L28).

[Comment] The result of Zdr from FIT has some characteristics of the observed Zdr but not that from KD. The Kdp from both FIT and KD differ quite significantly from the observation. Can you speculate why the Kdp is so poorly represented?

[Response] Since ZDR of our results are much poorer than KDP, we assumed that you were confused between ZDR and KDP. We are doubting some points to be modified in our method, for instance, axis ratio, quality control, and initial conditions, but no investigation has been done yet. This is one of future issues (P9L32-P10L2).

[Comment] From the Eq (19), Qr and Kdp have a quite simple relationship but why Zh is rather reasonable but not Kdp?

[Response] In KD, we can see quite small values of KDP (Fig. 2f), but good agreement with OBS in its horizontal distribution, while Zh looks better than KDP as Dr. Sun pointed out. However, since some of erroneous convections are seen in Zh, it is difficult to determine which one is truly better. We added the following sentence to P9L3.

"In KD, we can see quite small values of KDP (Fig. 2f), but good agreement with OBS in its horizontal distribution, while Zh looks better than KDP"

---

## Author Comment (AC3) · 23 May 2018

Dear Dr. Soichiro Sugimoto,

The authors deeply appreciate you for your useful comments. Our sincere responses are shown as follows and will be presented in the final form of our manuscript.

Sincerely, Takuya Kawabata on behalf of the authors

[General comment] The results, however, show the low performance in a simple assimilation experiment. The authors have to revise this manuscript mainly the relevant section of this 4DVar assimilation experiment before acceptance for publication. I think the authors need to reconsider the experimental configurations with additional consideration of the results, especially on the performance of FIT and the errors found in

differential reflectivity field. My comments are addressed below.

[Response] First of all, we do not propose new operators for dual polarimetric radar data assimilation in this manuscript, while we implemented forward operators, which were investigated their accuracy in Kawabata et al. (2018), and developed their TL and AD codes in the present study.

The first goal of our development of the operators was to obtain reasonable distribution of reflectivity (Zh) comparable to existing single polarimetric radar data assimilations. Since we thought to achieve the goal as illustrated in the examples (Fig. 2) and we believe that there is no vital problem existed in the operators, we finished the first stage of the development and moved on to the scientific stage for getting better results and further exploring scientific issues. Therefore, the purpose of this manuscript is providing technical information on the structure of the operators and their first examples, even though their performances were still poor.

To clarify abovementioned point, we modified the first sentence of the last paragraph of the introduction to "The scope of this paper is to provide the technical information on the observational operators and some evaluation results to help the users understand theoretical and practical aspects of the operators".

[Comment] 1. The authors had better mention clearly the range of application in abstract and summary. For example, operators can be applicable to C-band radar data (possibly S-band radar using findings of previous works). Another work, however, is needed to perform a statistical fitting of results simulated by a numerical radar simulator in applying to a radar system with a shorter wavelength (e.g., X-band radar). Besides, a dataset of beam-filling (effected by the ground) is required. In terms of a mesoscale model, the use of a two moment microphysical scheme is assumed.

[Response] We added the following sentences to mention applicable frequencies in the manuscript. With respect to the beam filling data and the two-moment scheme, we already mentioned them in Section 2.11 and Section 3 of the original manuscript,

respectively. "FIT is also applicable for X- and S-bands by replacing the coefficients. Although we already prepared the coefficients for all bands in the source codes, the users should carefully investigate their validity." (P5L9-10)

"Note that Eq. (19) is applicable for not only C-band but also X- and S-bands by putting their frequencies in f." (P5L15-16)

"Both of FIT and KD are applicable for not only C-band but also X- and S-bands." (P9L19-20)

[Comment] 2. The descriptions in Section 2.1 are quite similar to the description found in the previous work of Kawabata et al. (2018). The authors should explain the essence of FIT concisely for the readers to understand that the forward operators have been already proposed in another work. Please revise Section 2.1 carefully avoid double posting. This revise may be reflected to title.

[Response] Since the scope of this manuscript is providing the technical information on the operators, we decided to describe equations again. For avoiding the confusion for double posting, we replace the word "developed" (P1L9) with "implemented" in the abstract and added the following sentence to the last of Section 2.1.2:

"Eqs. (4)-(19) follow Kawabata et al. (2018), and we put the equations with different order in this manuscript for the readers' convenience to understand the flow of implementations of the forward, tangent linear and adjoint codes."

[Comment] 3. The authors have to describe experimental configurations (Section 4.2) in detail, including the domain, the grid spacing of the mesoscale model used, 4DVar timeline, radar data configuration (e.g., resolutions, the number of elevation angles) at least. Which mesoscale model is used, WRF or NHM? In terms of timeline, are several PPI data in one volume scan assimilated at the precise scanning timing during assimilation window? How about a method for preparing the background error?

[Response] We added the following sentences in Section 4.2:

"The horizontal resolution of NHM-4DVAR was 2 km and the length of assimilation window was 5 min, eleven PPI data from 0.5° to 4.8° elevations with the azimuth resolution at 0.7° and the range resolution of 150 m were assimilated at exact observation time as far as the time interval of NHM-4DVAR (10 s in this case) permits. The background errors were described in Kawabata et al. (2007) and (2011)."

"which displays the whole assimilation domain." (P9L4)

"The second one was done using WRF 3DVAR with actual radar data from the DWD radar network (Helmert et al. 2014) for the same case with "Case 1" described in Kawabata et al. (2018). The horizontal resolution of WRF 3DVAR was 2 km, and polarimetric parameters and rain water content in single PPI data by Offenthal radar was assimilated (see Kawabata et al. 2018 for detailed information on the observation). The background errors were calculated with ensemble simulations by WRF initialized by ECMWF analysis using the "gen_be" tool compiled in WRFDA. Observational errors were the same with the first case. From the increments of polarimetric parameters (Fig. 3), although quite small impacts are seen, similar patterns are recognized in both methods and larger impact of Zh and ZDR were produced in FIT and KD, respectively."

[Comment] 4. Observational errors are quite large, and, especially, the error for horizontal radar reflectivity seems to be unrealistic. A smaller error of radar reflectivity should be used. The authors may show the data to support the set-up of errors. The sensitivity of errors to the results should be discussed.

[Response] The values of the observational errors were determined after statistical investigations (Kawabata et al. 2018) but they were adopted conservatively (larger than the statistics). With respect to reflectivity, the value of 15 dBZ is the same with Kawabata et al. (2011). The sensitivity of the errors will be examined over the scientific explores. We added the following phrase at P8L21.

"which were determined after the statistical examination (Kawabata et al. 2018),"

How to estimate observational errors is one of the most important subjects in DA. The autocovariance method (see Appendix of Wulfmeyer et al. 2016) is more sophisticated than the try & error method and would be applied for radar data. We added the following phrase in Summary.

"it is necessary to estimate more appropriate observational errors (e.g., Wulfmeyer et al. 2016)." (P10L2)

[Comment] 5. Why not radial velocities (rv) assimilated in case of KD? Anyway, hail is associated with the event investigated. How does the authors consider the fall speed of hydrometeor in assimilating rv data?

[Response] We are sorry about the confusion on the rv assimilation. This was assimilated in KD as well. We modified the relevant sentence accordingly. Regarding the calculation method of rv, we used the same method with Sun and Crook (1997). We added the following sentence at the last of Section 4.

"In both cases, the radial velocity data were assimilated as the same method with Sun and Crook (1997)."

[Comment] 6. In Figure 2, non-zero differential reflectivity (Zdr) is calculated in non-stormy areas of retrieved and background fields, regardless of the type of operator. A possible reason regarding the axis ratio does not make sense to me. The authors should mention logically the reasons together with plotting smaller radar reflectivity Zh (< 15 dBZ). Assimilation of radar data (Zh, Zdr, Kdp) with quite weak echoes (e.g., clear air echo) is not appropriate in this framework. In operators, quite small Qr can lead to huge contribution in the perturbations of lamda, N0, and Zh (Zv). Therefore, the use of the minimum thresholds for Zh and Qr may remedy the low performance for the retrieval of Zdr, if the authors does not set the thresholds.

[Response] Thank you for your invaluable suggestion on improving the operators. Yes. It is necessary to polish up the quality control (QC) in the operators, although the

same QCs are applied as described in Kawabata et al. (2018). We think there is no QC applicable for any radar site, any DA system, and (could be) any case. Thus, we have to improve continuously the QCs. In addition, one of the candidates for the low performance is the axis ratio model as you mentioned. These are the future issues to be investigated. We mention these issues in the last paragraph of Summary.

"Furthermore, it is necessary to improve quality controls (QC) for polarimetric parameters, although the same QCs were applied as described in Kawabata et al. (2018), and the impact of axis ratio (Eq. (8)) and observational errors on assimilations will be investigated."

[Comment] 7. Although FIT is theoretically more precise than KD, FIT shows the lower performance than KD. The authors jump to conclusions too quickly by regarding the nonlinearity in FIT as the low performance. In the background Zh and Kdp, the mesoscale model cannot resolve convections at all. One possible situation is that it is too dry in the background water vapor field. If so, I guess the adjustment of humidity is needed before assimilation to retrieve larger Zh and Kdp.

[Response] We agree with you. Actually, the initial conditions of these cases were quite poor. It is necessary to enhance the quality of the initial conditions, for instance, by using ensemble Kalman filter systems or downscaling the more appropriate analysis fields. We mentioned this matter in Summary.

"It would be possible to overcome the weaknesses of the Zh distributions in FIT and FG through assimilation–forecast cycles and/or by adding other types of observation data, such as conventional observations, Doppler (water vapour) lidar data, and water vapour data observed by GNSS."

Specific and minor comments: [Comment] 1. (Page 2) Why do the authors address on quantitative precipitation estimation (QPE)? I think QPE is out of the main topic of this manuscript.

[Figure]

[Response] As shown in Bauer et al. (2015), high resolution DA systems with rapid update cycle is able to produce rainfall distributions comparable to actual observations. We added the following words at P2L9.

", which was discussed in Bauer et al. (2015) in the context of a data assimilation with high resolution and rapid update cycle,"

[Comment] 2. (Page 2 Line7) Change "is" to "are".

[Response] We got. Thank you.

[Comment] 3. (Pages 2 and 3) Operators proposed consider the relations between variables concerning rain water and radar observables. I feel something wrong with the mention of "cloud water". Can C-band radar observe cloud water?

[Response] No. We removed all "cloud water" from the manuscript.

[Comment] 3. (Pages 2 and 3) Please check if the WRFDA "(WRF-Var" mentioned in the manuscript) deals with the "perturbation" of rainwater mixing ratio, not dealing with "the total part".

[Response] The prognostic variable in WRF is rainwater itself, not the total water, if you mentioned this point.

[Comment] 4. (Page 4 Line 6) Change "proportional" to "polynomial".

[Response] Modified.

[Comment] 5. (Page 8 Line 17) Essentially, is Kdp assimilate in KD?

[Response] Yes. We added "derived from KDP" after Qrain.

[Comment] 6. (Page 9 Lines 8 & 9) The performance found in a simple assimilation test is far from a successful level.

[Response] Since these statements are from the comparison with the first guess field (FG), which catches no rain it is clear that the results are better than FG. However,

since we agree with you that ZDR from both methods are poor, we will add "except for ZDR" at the end of the sentence.

---

## Referee Report (RR1)

Minor comments:

1. (Page 8, Line 22) If the observation error of $Z_h$ is larger than the error investigated in Kawabata et al. 2018, the authors should address that the value of 15 dBZ is conservative one. Why do the authors use such a larger error?

2. (Page 8, Line 22) Are the observational errors homogeneous spatially or different depending on the distance from the radar site?

3. (Page 1, Line 9) The authors revised so that the word "implemented" is used twice in this sentence. The sentence may need to be polished.

4. (Page 3, Lines 8-15) The authors revised to remove the phrase "cloud water". There are words to be expressed as singular form.

5. (Page 11, Line 14) Start a new line for Brandes et al. (2002).

6. (Page 11, Line 29) Remove this line.

7. (Pages 11-13) Use italic for journal titles.

---

## Author Response (AR2)

Dear Dr. Sugimoto,

Thank you for valuable comments. We modified the manuscript after your comments. Point-by-point responses are shown below.

Sincerely,
Takuya Kawabata on behalf of authors.

1. (Page 8, Line 22) If the observation error of Zh is larger than the error investigated in Kawabata et al. 2018, the authors should address that the value of 15 dBZ is conservative one. Why do the authors use such a larger error?

The reason is that "15 dBZ" is the same with the previous study by Kawabata et al. (2011). We added the phrase "which is the same with Kawabata et al. (2011)" after "15.0 dBZ" (P8L22).

2. (Page 8, Line 22) Are the observational errors homogeneous spatially or different depending on the distance from the radar site?

These errors are homogeneous in space, which means observational error covariances are diagonal. We added this sentence at P8L23.

3. (Page 1, Line 9) The authors revised so that the word "implemented" is used twice in this sentence. The sentence may need to be polished.

Modified. Thank you.

4. (Page 3, Lines 8-15) The authors revised to remove the phrase "cloud water". There are words to be expressed as singular form.

Modified. Thank you.

5. (Page 11, Line 14) Start a new line for Brandes et al. (2002).

Modified. Thank you.

6. (Page 11, Line 29) Remove this line.

Modified. Thank you.

7. (Pages 11-13) Use italic for journal titles.

Modified. Thank you.